# Sesquiterpenes from the Fungus *Antrodiella albocinnamomea* with Cytotoxicity and Antibacterial Activity

**DOI:** 10.3390/jof9050521

**Published:** 2023-04-28

**Authors:** Jinlei Ning, Feng Wu, Jikai Liu, Juan He, Tao Feng

**Affiliations:** School of Pharmaceutical Sciences, South-Central Minzu University, Wuhan 430074, China; jlning026@163.com (J.N.); wfeng828@163.com (F.W.); jkliu@mail.kib.ac.cn (J.L.)

**Keywords:** *Antrodiella albocinnamomea*, sesquiterpene, cytotoxicity, antibacterial activity

## Abstract

Eight new sesquiterpenes, namely, albocinnamins A−H (**1**−**8**), along with two known ones (**9** and **10**), have been isolated from the fungus *Antrodiella albocinnamomea*. Compound **1** possesses a new backbone that might be derived from cadinane-type sesquiterpene. Structures of the new compounds were elucidated by detailed spectroscopic data analysis, single-crystal X-ray diffraction, and ECD calculations. Compounds **1a** and **1b** showed cytotoxicity against SW480 and MCF-7 cells, with IC_50_ values ranging from 19.3 to 33.3 μM, while compound **2** displayed cytotoxicity against the HL-60 cell with an IC_50_ value of 12.3 μM. In addition, compounds **5** and **6** exhibited antibacterial activity against *Staphylococcus aureus* with MIC values of 64 and 64 µg/mL, respectively.

## 1. Introduction

*Antrodiella albocinnamomea* is a white-rot basidiomycetous fungus belonging to Basidiomycota, which is widely distributed in temperate to subtropical areas of China [1]. Previous studies show that *A. albocinnamomea* is highly productive for bioactive sesquiterpenes, including cadinane, triquinane, chamigrane, humulane, and gymonmitrane classes [2,3,4,5,6,7,8]. Representative sesquiterpenes are antroalbocin A, antroxazole A, antrodillin, and antroalbol H. Antroalbocin A is a novel bridged tricyclic sesquiterpene with antibacterial activity against *Staphylococcus aureus* [9]. Antroxazole A is an interesting chamigrane dimer containing an oxazole moiety, showing selective inhibition on LPS-induced B lymphocyte cell proliferation [10]. Antrodillin is a triquinane sesquiterpene derivative that also showed immunosuppressive activity [11]. Antroalbol H is a chamigrane sesquiterpene that has potential anti-diabetic activity [12]. Such rich sesquiterpene resources prompted us to carry out further research on this fungus. As part of our long-term research into the chemical composition of fungi, we conducted the secondary metabolites on the cultural broth of the fungus *A. albocinnamomea* in rice medium. Herein, the isolation, structural elucidation, and antibacterial activity of these isolates (Figure 1) are reported.

## 2. Materials and Methods

### 2.1. General Expriment Procedures

Melting points were measured on an X-4 micro melting point apparatus. Optical rotations were acquired using a Rudolph Autopol IV polarimeter. UV and CD spectra were recorded on a UH5300 UV-VIS Double Beam Spectrophotometer and an Applied Photophysics Chirascan-Plus spectrometer. IR spectra were conducted on a Shimadzu Fourier transform infrared spectrometer with KBr pellets. 1D and 2D NMR spectra were recorded on a Bruker Advance III 600 spectrometer using TMS as a ternal standard. Chemical shifts (*δ*) are reported in parts per million (ppm). HRESIMS data were obtained on a Thermo Scientific Q Exactive Orbitrap MS system. X-ray crystallographic analysis was conducted on the BRUKER D8 QUEST. Colum chromatography (CC) was carried out using silica gel (200–300 and 500–800 mesh, Qingdao Marine Chemical Ltd., Qingdao, China), RP-18 gel (20–45 µ, Fuji Silysia Chemical Ltd., Aichi, Japan), and Sephadex LH-20 (Pharmacia Fine Chemical Co., Ltd., Uppsala, Sweden). Medium Pressure Liquid Chromatography (MPLC) was performed on Biotage SP1 equipment, and columns were packed with RP-18 gel. High performance liquid chromatography (HPLC) was performed on Agilent 1260 system, equipped with DAD detector, Agilent ZORBAX SB-C18 column (5 µm, 4.6 × 150 mm) and Agilent XDB-C18 column (5 µm, 9.4 × 150 mm or 21.2 × 150 mm). The results were monitored by thin-layer chromatography (TLC). All solvents used were of analytical grade.

### 2.2. Fungal Material

The fungus *A. albocinnamomea* was collected from rotting poplar trees in Changbai Mountain Nature Reserve, Jilin Province of China, on 20 October 2009. It was identified by Professor Yu-Cheng Dai (Beijing Forestry University). The fungal specimen (CGBWSHF00182-4) has been deposited at the School of Pharmaceutical Sciences, South-Central Minzu University, China. The strain was cultured on plates of potato dextrose agar (PDA) medium at 25 °C for 6 days. After that, several pieces of mycelium were inoculated into rice culture medium (100 g of rice, 100 mL of water, in each 500 mL culture bottle). A total of 200 bottles were incubated fixedly at 25 °C for 40 days in a dark place.

### 2.3. Extraction and Isoation

The fermented material was extracted five times with absolute methanol. The extract was dissolved in water and EtOAc, and extracted four times with EtOAc to yield 124.0 g. As shown in Figure 1, the crude extract was separated into 9 fractions (A−I) using a CC over silica gel column (80–100 mesh) with a solvent system of CH_2_Cl_2_−MeOH (from 100:0 to 0:100). Fr. E (16.0 g) was separated by MPLC over RP-18 silica gel eluted with MeOH/H_2_O (from 10:90 to 100:0) to obtain 11 fractions (E1−E11). Fr. E4 and Fr. E5 were both isolated with a silica gel column with a step gradient of CH_2_Cl_2_−MeOH (from 100:1 to 0:100) to afford 10 parts (E4-1−E4-10 and E5-1−E5-10). Fr. E4-5 was fractionated by Sephadex LH-20 CC eluted with acetone into five subfractions (E4-5-1−E4-5-5). Fr. 4-5-2 was purified by prep-HPLC to yield compound **4** (2.6 mg). Fr. E4-5-4 was subjected to Sephadex LH-20 CC eluted with MeOH and further purified by prep-HPLC to yield compound **1** (3.0 mg). Fr. E5-6 was prepared with HPLC to yield compound **9** (6.0 mg). Fr. E5-7 was separated into 8 parts (Fr. E5-7-1−5-7-8) using RP-18 silica gel eluted with MeOH/H_2_O (from 20:80 to 0:100). Fr. E5-7-5 was purified by prep-HPLC to yield compound **10** (5.5 mg). Fr. F (10.0 g) was first fractionated by MPLC over RP-18 eluted with MeOH/H_2_O (from 20:80 to 0:100) to give 14 subfractions (F1−F14). Fr F4 was isolated with Sephadex LH-20 CC eluted with CH_2_Cl_2_/MeOH (1:1) to afford 8 parts (F4-1−F4-8). Further purification of Fr. 4-3 with prep-HPLC gave compounds **3** (2.2 mg) and **2** (1.6 mg). Fr. F6 was separated into 7 subfractions (F6-1−F6-7) by a silica gel column of CH_2_Cl_2_/MeOH (from 50:1 to 0:100). Fr. F6-4 was purified by Sephadex LH-20 CC eluted with MeOH to obtain 5 parts (F6-4-1−F6-4-5). Fr. F6-4-2 was then further purified via prep-HPLC to give compounds **5** (9.0 mg) and **7** (1.6 mg). Fr. F7 was separated into 9 subfractions (F7-1−F7-9) by a silica gel column of petroleum ether/acetone (from 30:1 to 0:1). Fr. F7-3 was prepared with HPLC to yield compound **8** (3.4 mg). Fr. F7-5 was isolated with Sephadex LH-20 CC eluted with acetone to afford 6 parts (F7-5-1−F7-5-6). Fr. 7-5-3 was purified by prep-HPLC to yield compound **6** (8.0 mg).

*Albocinnamin A* (**1a**/**1b**)*:* colorless crystals (H_2_O); mp 182.5–186.8 °C; **1a**: [*α*]^25^_D_ + 46.9 (*c* 0.50, MeOH), **1b**: [*α*]^25^_D_ − 22.5 (*c* 0.50, MeOH); UV (H_2_O) *λ*_max_ (log *ε*) 215 (3.78), 230 (3.68), 290 (3.22) nm; ^1^H (methanol-*d*_4_, 600 MHz) and ^13^C NMR (methanol-*d*_4_, 150 MHz) data, see Table 1; HRESIMS *m*/*z* 263.12769 [M + H] ^+^ (calcd for C_15_H_20_O_3_, 263.12779).

*Albocinnamin B* (**2**)*:* colorless oil; [*α*]^23.6^_D_ − 23.6 (*c* 0.5, MeOH); UV (MeOH) *λ*_max_ (log *ε*) 235 (3.49) nm; ^1^H (600 MHz) and ^13^C NMR (150 MHz) data (methanol-*d*_4_), see Table 1; HRESIMS *m*/*z* 235.16927 [M + H] ^+^ (calcd. for C_15_H_23_O_3_, 235.16926).

*Albocinnamin C* (**3**)*:* yellow powder; [*α*]^23.6^_D_ + 9.6 (*c* 0.5, MeOH); UV (MeOH) *λ*_max_ (log *ε*) 235 (3.59) nm; ^1^H (600 MHz) and ^13^C NMR (150 MHz) data (methanol-*d*_4_), see Table 1; HRESIMS *m*/*z* 287.16153 [M + Na] ^+^ (calcd. for C_16_H_24_NaO_3_, 287.16177).

*Albocinnamin D* (**4**)*:* colorless oil; [*α*]^23.6^_D_ + 13.6 (*c* 0.5, MeOH); UV (MeOH) *λ*_max_ (log *ε*) 235 (3.86) nm; ^1^H (600 MHz) and ^13^C NMR (150 MHz) data (methanol-*d*_4_), see Table 1; HRESIMS *m*/*z* 251.16422 [M + H] ^+^ (calcd. for C_15_H_23_O_3_, 251.16417).

*Albocinnamin E* (**5**)*:* colorless crystals (H_2_O); mp 234–236 °C; [*α*]^27^_D_ − 108.7 (*c* 0.5 MeOH); UV (MeOH) *λ*_max_ (log *ε*) 235 (3.98) nm; IR (KBr) *ν*_max_ 3329, 2960, 1697, 1649, 1379, 1022 cm^−1^; ^1^H (600 MHz) and ^13^C NMR (150 MHz) data (methanol-*d*_4_), see Table 2; HRESIMS *m*/*z* 251.16414 [M + H] ^+^ (calcd. for C_15_H_23_O_3_, 251.16417).

*Albocinnamin F* (**6**)*:* colorless oil (H_2_O); [*α*]^27^_D_ − 53.2 (*c* 0.5 MeOH); UV (MeOH) *λ*_max_ (log *ε*) 245 (3.83) nm; ^1^H (600 MHz) and ^13^C NMR (150 MHz) data (DMSO-*d*_6_), see Table 2; HRESIMS *m*/*z* 267.15892 [M + H] ^+^ (calcd. for C_15_H_23_O_4_, 267.15909).

*Albocinnamin G* (**7**)*:* colorless oil (H_2_O); [*α*]^27^_D_ + 166.7 (*c* 0.5, MeOH); UV (MeOH) *λ*_max_ (log *ε*) 270 (3.20) nm; ^1^H (600 MHz) and ^13^C NMR (150 MHz) data (methanol-*d*_4_), see Table 2; HRESIMS *m*/*z* 249.14854 [M + H] ^+^ (calcd. for C_15_H_21_O_3_, 249.14852).

*Albocinnamin H* (**8**)*:* colorless oil; [*α*]^27^_D_ + 44.4 (*c* 0.5, MeOH); UV (MeOH) *λ*_max_ (log *ε*) 230 (3.47) nm; ^1^H (600 MHz) and ^13^C NMR (150 MHz) data (methanol-*d*_4_), see Table 2; HRESIMS *m*/*z* 305.13573 [M + Na] ^+^ (calcd. for C_15_H_22_NaO_5_, 305.13594).

### 2.4. ECD Calculations

The *Gaussian 16* program package was used for the calculations of the ECD spectra of **1**−**3**, **6**−**7**. The stable conformers subjected to ECD calculation were optimized using the time-dependent density functional theory (TDDFT) method at the B3LYP/6-311G (d, p) level of theory [13,14]. The ECD curves were extracted by SpecDis 1.60 and weighted by Boltzmann distribution after UV correction [15]. For details, see the Appendix A below.

### 2.5. X-Ray Crystallographic Analysis

Single crystals of compounds **1**(**1a**/**1b**) and **5** were obtained from MeOH and H_2_O, and all single crystals were collected by a Bruker D8 QUEST diffractometer, which was equipped with Cu-Kα radiation (λ 1.54178 Å). The structure was solved with ShelXT, using direct methods and refined with ShelXT using least square minimization. Crystallographic data for compounds **1** and **5** have been deposited at the Cambridge Crystallographic Data Centre (CCDC number for **1**: 2253029, and **5**: 2253030).

*X-ray crystallographic data for ***1** (**1a** and **1b**)*:* C_15_H_18_O_4_, (*M* = 262.29 g/mol): monoclinic, space group Pbca, *a* = 11.1153(5) Å, *b* = 13.4703(6) Å, *c* = 18.6151(8) Å, *α* = 90°, *β* = 90°, *γ* = 90°, *V* = 2787.2(2) Å^3^, *Z* = 8, *T* = 100(2) K, *μ*(Cu Kα) = 0.739 mm^−1^, *Dcalc* = 1.250 mg/m^3^, 22,765 reflections measured, 2749 independent reflections (*R_int_* = 0.0608). The final *R*_1_ was 0.0347 (*I* > 2*σ*(*I*)), and *wR*(*F*^2^) was 0.0923 (*I* > 2*σ*(*I*)). The final *R*_1_ was 0.0456 (all data), and *wR*(*F*^2^) was 0.0962 (all data).

*X-ray crystallographic data for **5**:* C_15_H_22_O_3_, (*M* = 250.32 g/mol): monoclinic, space group P2**_1_**2**_1_**2**_1_**, *a* = 7.7864(4) Å, *b* = 9.0991(4) Å, *c* = 19.0099(9) Å, *α* = 90°, *β* = 90°, *γ* = 90°, *V* = 1346.84(11) Å^3^, *Z* = 4, *T* = 100(2) K, *μ*(Cu Kα) = 0.676 mm^−1^, *D_calc_* = 1.235 mg/m^3^, 12,794 reflections measured, 2639 independent reflections (*R_int_* = 0.0419). The final *R*_1_ was 0.0309 (*I* > 2*σ*(*I*)), and *wR*(*F*^2^) was 0.0806 (*I* > 2*σ*(*I*)). The final *R_1_* was 0.0312 (all data), and *wR*(*F*^2^) was 0.0808 (all data). Flack parameter = 0.01(5).

### 2.6. Antibacterial Assay

All compounds were subjected to minimal inhibitory concentration (MIC) tests against two species of bacteria (*S. aureus* and *Mycobacterium tuberculosis*). Both bacteria were purchased from China General Microbiological Culture Collection Center (CGMCC). All strains were cultured in Mueller Hinton broth (MHB) (Guangdong Huankai Microbial Sci. &Tech. Co., Ltd., Guangzhou, China) at 37 °C. A sample of each culture was then diluted 40-fold in fresh MHB broth and incubated at 37 °C with shaking (200 rpm) for 2.5 h [16]. The resultant mid-log phase cultures were diluted to a concentration of 5 × 10^5^ CFU/mL, and then 50 mL was added to each well of the compound-containing plates, giving a final compound concentration range of 128 or 50 mg/mL. The plates were observed after 24 h incubation at 37 °C [17]. Inhibition rates were determined using photometry at OD_625 nm_. Rifampicin was used as the positive control (MIC < 2.5 µg/mL).

### 2.7. Cytotoxicity Assay

All compounds were assessed for their cytotoxicity toward the human promyelocytic leukemia (HL-60), colon cancer (SW480), and breast cancer (MCF-7) cell lines. All the cells were seeded into 96-well plates containing DMEM or RPMI1640 medium with 10% FBS under a 5% CO_2_ atmosphere at 37 °C. The assays were performed by the MTS method according to the manufacturer’s instructions [18]. Briefly, the isolated compounds dissolved in dimethyl sulfoxide (DMSO) and were then diluted with culture media to produce difference concentrations (40, 20, 10, 5, 2.5, 1.25, 0.625 µM). After incubation for 24 h, various levels of compounds were added to each well and incubated for 48 h. A total of 100 µL of culture media and 20 µL of MTS solution were added, which incubated for 3 h at 37 °C [19]. The absorbance of each well was measured at 490 nm using the Multi-Mode microplate reader. Paclitaxel was used as a positive control, and the concentrations for paclitaxel were 0.5, 0.25, 0.125, 0.0625, 0.03125, 0.015625, and 0.0078125 µM (IC_50_ < 0.08 µM).

## 3. Results and Discussion

Compound **1** was isolated as colorless crystals. Its molecular formula was determined as C_15_H_19_O_3_ by HRESIMS (measured at *m*/*z* 263.12769 [M + H]^+^; calcd for C_15_H_20_O_3_, 263.12779), which accounted for seven double-bond equivalents. The ^1^H NMR and ^13^C NMR spectrum (Table 1) showed 15 carbon signals, including four CH_3_, five CH, and six non-protonated carbons. Primary analysis of these data indicated that **1** had a benzene group and two carbonyl carbons. The ^1^H-^1^H COSY data revealed three fragments, as shown in Figure 2. Based on this, the HMBC data revealed the planar structure of **1** (Figure 2). At first, the HMBC correlations from H_3_-14 to C-1 and from H-11 to C-3, C-4, C-5 indicated one methyl group placed at C-1 and an isopropyl group placed at C-4 of the benzene group, respectively. In addition, the HMBC correlations from H-6 to C-9 (*δ*_C_ 173.3) suggested a γ-lactone fused with the benzene group. Finally, the HMBC correlations from H_3_-15, H-6 and H-7 to C-8 (*δ*_C_ 176.9) suggested a carboxyl group of C-8, in connection with C-7. Hence, the planar structure of **1** was established as an aromatic sesquiterpene with a novel backbone. A single crystal X-ray diffraction experiment was performed (Figure 3), and the result confirmed the planar structure as given above. In addition, the data revealed the relative configuration of **1** and suggested that **1** should be a racemate. Therefore, compound **1** was separated into two pure enantiomers by chiral-phase HPLC (Appendix A), and the absolute configurations were determined by comparing the calculated and experimental ECD spectra (**1a**/**1b**, Figure 4). This experiment enabled **1a** and **1b** to be determined as (+)-(6*S*,7*R*) and (−)-(6*R*,7*S*), respectively. Consequently, the structure of **1** was characterized and trivially named as (±)-albocinnamin A.

Compound **2** was isolated as a colorless oil. Its molecular formula was determined as C_15_H_22_O_2_ by HRESIMS (measured at *m*/*z* 235.16927 [M + H]^+^; calcd for C_15_H_23_O_2_ 235.16926), which accounted for five degrees of unsaturation. The ^1^H and ^13^C NMR data (Table 1) revealed four CH_3_, one CH_2_, seven CH, and three non-protonated carbons. Of them, data at *δ*_C_ 152.1 (d, C-6), 135.1 (s, C-7) and 201.0 (s, C-8) indicated the presence of an *α*,*β*-unsaturated keto moiety, while one signal at *δ*_C_ 69.3 (d, C-3) suggested one OH group. The ^1^H-^1^H COSY correlations disclosed a long link, as shown in Figure 2, which established the OH position at C-3 and revealed the *α*,*β*-unsaturated keto moiety to be 6,7-en-8-one. Based on this, the HMBC correlations from H-14 to C-1, C-2, and C-10 constructed a six-membered ring. In addition, the HMBC correlations from H-9 to C-8 and C-7 established another six-membered ring. Therefore, compound **2** was assigned as a bicyclic cadinane sesquiterpene [20]. In the ROESY spectrum (Figure 5), observed cross peaks of H-3/H-5, H-12 and H-5/H-10 indicated that H-3, H-5 and H-10 were in the same orientation, and H-4 was in the other orientation. There, correlations revealed the relative configuration of **2** (3*S*,4*R*,5*S*,6*R* or 3*R*,4*S*,5*R*,6*S*). Finally, the absolute configurations were determined by comparing the calculated and experimental ECD spectra (Figure 4). Hence, the structure of **2** was identified and trivially named as albocinnamin B.

Compound **3** was isolated as a yellow solid. Its molecular formula was determined as C_16_H_24_O_3_ by HRESIMS (measured at *m*/*z* 287.16153 [M + Na]^+^; calcd for C_16_H_24_NaO_3_ 287.16177). The ^1^H and ^13^C NMR data (Table 1) showed similarities to those of **2,** except for the substitutions at C-3 and C-9. There is a hydroxyl group at C-3 of **2**, while C-3 of **3** is a methoxy group. Finally, the signal at *δ*_C_ 76.0 (d, C-9) indicated a OH group placed at C-9. These changes were substantiated by the HMBC correlation from H-16 to C-3, and the ^1^H-^1^H COSY correlation from H-9 to H-10. In the ROESY spectrum (Figure 5), observed cross peaks of H-3/H-5, H-12, H-5/H-10 and H-4/H-9 indicated that H-3, H-5 and H-10 were in the same orientation, and that H-4 and H-9 were in the same orientation. Finally, the absolute configuration of **3** was determined by ECD calculations (Figure 4). Consequently, the structure of **3** was identified and trivially named as albocinnamin C.

Compound **4** was isolated as a colorless oil. Its molecular formula was determined as C_15_H_22_O_3_ by HRESIMS (measured at *m*/*z* 251.16422 [M + H]^+^; calcd for C_15_H_23_O_3_ 251.16417). Primary analysis of 1D and 2D data (Table 1) was similar to those of **3**. The difference was that one carbonyl carbon (*δ*_C_ 204.5) placed at C-3 and one OH group placed at C-8. This conclusion was supported by the HMBC correlations from H-4, H-2 to C-3, and from H-15 to C-6, C-7, and C-8. In the ROESY spectrum, cross peaks of H-5/H-10, H-8, and H-4/H-9 were observed (Figure 2). Considering the homology of biological sources, the absolute configuration of **4** should be the same as **3**. Hence, the structure of **4** was identified and trivially named as albocinnamin D.

Compound **5** was isolated as colorless crystals. Its molecular formula was determined as C_15_H_22_O_3_ by HRESIMS (measured at *m*/*z* 251.16414 [M + H]^+^; calcd for C_15_H_23_O_3_ 251.16417), which accounted for five double-bond equivalents. The ^1^H NMR and ^13^C NMR spectrum (Table 2) of compound **5** showed signals for 15 carbons, including three CH_3_, four CH_2_, three CH, and five non-protonated carbons. Primary analysis of 1D and 2D data showed that **5** was similar to cocumin F [21]. The major difference between **5** and cocumin F is that the oxygenated methylene in **5** replaced the methyl of C-12 in cocumin F, which was confirmed by the HMBC correlations from H-12 to C-9, C-10 and C-11 (Figure 2). In the ROESY spectrum (Figure 5), the cross peaks of H-1/H-8, H-1/H-13, H-7/H-8, H-8/H-13 were observed, and not cross peaks of H_3_-14/H-1, H-7 and H-8. The planar structure of **5** was indicated. Finally, a single-crystal X-ray diffraction not only confirmed the planar structure, but also established the absolute configuration of **5** (Figure 3), named as albocinnamin E.

Compound **6** was isolated as a colorless oil. Its molecular formula was determined as C_15_H_22_O_4_ by HRESIMS (measured at *m*/*z* 267.15892 [M + H]^+^; calcd for C_15_H_23_O_4_ 267.15909). The 1D and 2D NMR data (Table 2) showed that it was extremely similar with those of **5**, except for the presence of an additional OH group, which was located at C-1, as confirmed by the HMBC correlations from H-8 and H-11 to C-1. In the ROESY spectrum (Figure 5), observed cross peaks of 1-OH/H-7, H-8, and without cross peaks of H_3_-14/H-7, H-8, indicating that 1-OH, H-7, and H-8 were in the same orientation and H_3_-14 was in the other orientation. Finally, the absolute configuration of **6** was confirmed by ECD calculations (Figure 4). Hence, the structure of **6** was identified and trivially named as albocinnamin F.

Compound **7** was isolated as a colorless oil. Its molecular formula was determined as C_15_H_20_O_3_ by HRESIMS (measured at *m*/*z* 249.14854 [M + H]^+^; calcd for C_15_H_21_O_3_ 249.14852), accounting for six degrees of unsaturation. The 1D data (Table 2) displayed high similarity to those of **5**, except for the signals at *δ*_C_ 143.6 and *δ*_C_ 135.7, which suggested one double bond between C-1 and C-11 of **7**. This assumption was supported by the HMBC correlations from H-8 to C-1, C-11 (Figure 2). Detailed analysis of 2D NMR data suggested that other parts of **7** were identical to those of **5**. The absolute configurations were determined by comparing the calculated and experimental ECD spectra (Figure 4). Therefore, the structure of **7** was established and trivially named as albocinnamin G.

Compound **8** was isolated as a colorless oil. Its molecular formula was determined as C_15_H_22_O_5_ by HRESIMS (measured at *m*/*z* 305.13573 [M + Na]^+^; calcd for C_15_H_22_NaO_5_ 305.13594), which accounted for five degrees of unsaturation. Primary analysis of the ^1^H and ^13^C NMR (Table 2) spectra of **8** showed its close resemblance to ochracine F,^15^ and the difference of **8** was the presence of one OH group placed at C-9, as supported by the ^1^H-^1^H COSY correlation from H-9 to H-10 and its molecular weight. In the ROESY spectrum, observed cross peaks of H-9/H-14 indicated H-9 and H-14 in the same orientation (Figure 5). Considering the homology of biological sources, the absolute configuration of **8** was the same as that of ochracine F. Consequently, the structure of **8** was identified and trivially named as albocinnamin H.

In addition, two known compounds were identified as ochracine F (**9**) [22] and cerrenin C (**10**) [23] by comparison of their spectroscopic data to the reported data in the literature. Structurally, compound **1** possessed a new backbone, and it may be derived from a precursor of the cadinane sesquiterpene **2** via a key Baeyer−Villiger oxidation. After further hydrolysis, oxidation, dehydration, and aromatization, a novel product **1** was finally built (Figure 2).

Previous studies have proved that sesquiterpenes in *A. albocinnamomea* have cytotoxicity and antibacterial activity [9]. Therefore, all compounds were tested for their cytotoxicity activity against three human cancer cell lines (HL-60, SW480 and MCF-7) and for their antibacterial activity against *S. aureus* and *M. tuberculosis*. As a result, compounds **1a** and **1b** showed moderate cytotoxicity against SW480 and MCF-7 cells with IC_50_ values ranging from 19.3 to 33.3 μM, while compound **2** showed cytotoxicity against HL-60 cells with an IC_50_ value of 12.3 μM (Table 3). In addition, compounds **5** and **6** exhibited antibacterial activity against *S. aureus* with MIC values of 64 and 64 µg/mL, respectively.

## 4. Conclusions

A total of ten sesquiterpenes, including eight new ones, have been characterized from the fungus *A. albocinnamomea*. The new structures with absolute configurations were established by means of spectroscopic methods, single-crystal X-ray diffraction, and ECD calculations. Compounds **1a** and **1b** showed cytotoxicity against SW480 and MCF-7 cells, while compound **2** displayed cytotoxicity against HL-60 cell. In addition, compounds **5** and **6** exhibited antibacterial activity against *S. aureus*. This indicates that *A. albocinnamomea* is rich in sesquiterpenes, which have potential cytotoxicity and antibacterial application prospects.

## Data Availability

The data of this study are available from the corresponding author (T.F.).

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
