# Peer review of "Sesquiterpenes from the Fungus Antrodiella albocinnamomea with Cytotoxicity and Antibacterial Activity"

_jof, 2023, doi:10.3390/jof9050521_

Round 1

Reviewer 1 Report

The manuscript entitled:  Sesquiterpenes from the Fungus Antrodiella albocinnamomea with Cytotoxicity and Antibacterial Activity, presents interesting study in which 8 new Sesquiterpenes where isolated from the Fungus Antrodiella albocinnamomea. Yet the flowing points should be addressed before further steps:

Abstract

Missing value in (64 μg/mL,……, respectively)

Introduction

-The last 2 sentences: rewrite by avoiding mentioning your specific results and figures which should be left to the results and discussion parts.

Materials:

-It is advised to make a diagram showing the different stages and yield of extraction.

-What is the source of Staphylococcus aureus, Mycobacterium tuberculosis and the cancer cells

-Selectivity assay must be performed by testing the compounds against normal cells (i.e. MRC5 fibroblasts), to ensure that the compounds do not kill normal human cells.

-2.7: specify the compounds and paclitaxel concentrations used for the MTS assay rather than just mentioning ‘’various levels’’.

needs minor revsions

Author Response

Abstract

Missing value in (64 μg/mL,……, respectively)

Re. Dear reviewer, both compounds showed the same antibacterial level. We revised is as “compounds 5 and 6 exhibited antibacterial activity against Staphylococcus aureus with MIC values of 64 and 64 µg/mL, respectively”

Introduction

-The last 2 sentences: rewrite by avoiding mentioning your specific results and figures which should be left to the results and discussion parts.

Re. We have revised it. Deleted the words for key results.

Materials:

-It is advised to make a diagram showing the different stages and yield of extraction.

 Re. We added a diagram for this section.

-What is the source of Staphylococcus aureus, Mycobacterium tuberculosis and the cancer cells

 Re. The strains were purchased from China General Microbiological Culture Collection Center (CGMCC). We have added this information in the text.

-Selectivity assay must be performed by testing the compounds against normal cells (i.e. MRC5 fibroblasts), to ensure that the compounds do not kill normal human cells.

 Re. Dear reviewer, we did the test of the compounds on the mice normal cell RAW264.7. Compounds 1 and 2 showed no cytotoxicity to RAW264.7vat the concentration of 40 μM. Since it is not human normal cell, we didn’t add this information in the text. Currently, we have no human normal, we hope that we could finish this experiment in the future.

-2.7: specify the compounds and paclitaxel concentrations used for the MTS assay rather than just mentioning ‘’various levels’’.

 Re. We have specified the concentrations for compounds and control.

Reviewer 2 Report

This manuscript discusses the structure elucidation and biological evaluation of new sesquiterpenes isolated from Antrodiella albocinnamomea.

The experiments described in this paper seem to have been conducted with appropriate expertise. The writing is clear and the results are likely to be very interesting for readers.

After reading the manuscript, the following concern arose:

1. Figure 4: It would be helpful if the units for ?ε were more understandable.

My evaluation is that this manuscript requires very minor revisions before being ready for publication.

Author Response

After reading the manuscript, the following concern arose:

  1. Figure 4: It would be helpful if the units for ?ε were more understandable.

My evaluation is that this manuscript requires very minor revisions before being ready for publication.

Re. Dear reviewer, we checked the figure 4 and added the units for Δε. In addition, we have checked the manuscript carefully, making it more suitable for publication.

Reviewer 3 Report

The manuscript is good and well written. but some revisions should be done like

1. The introduction should be increased with more details

2. The aim of the work should be written in good style

3. The Dose response curves of the cytotoxic assays should be inserted

4. Antibacterial Assay!!!!!!. Where the results?????????

English is good but a minor revisions should be done

Author Response

  1. The introduction should be increased with more details

Re. Dear reviewer, we have studied for the chemical constituents of the fungus Antrodiella albocinnamomea for a long time, and previous papers have introduced enough information for this strain. Therefore, we did not introduce too much in this paper. Thank you.

  1. The aim of the work should be written in good style.

Re. We made simply revisions.

  1. The Dose response curves of the cytotoxic assays should be inserted

Re. The cytotoxic assay was achieved by Kunming Institute of Botany, CAS. It did not give dose curves. We will ask for this in the experiment for the future.

  1. Antibacterial Assay!!!!!!. Where the results?????????

Re. The results were given in the Abstracts and Results and discussion. Both compounds showed the same antibacterial activity with MIC of 64 μg/mL.

Round 2

Reviewer 1 Report

Thank you

Reviewer 3 Report

I think the manuscript can be accepted in the present format 

I think the manuscript can be accepted in the present format